# Quality Improvement and Shelf-Life Extension of Iced Nile Tilapia Fillets Using Natural Garlic Extract

Edgar Iván Jiménez-Ruíz [1], Víctor Manuel Ocaño-Higuera [2,*], Santiago Valdez-Hurtado [3,*], José Alberto Cruz-Guzmán [2], Cesar Benjamín Otero-León [2], Saúl Ruíz-Cruz [4], Alba Mery Garzón-García [5], Hebert Jair Barrales-Cureño [6], Dalila Fernanda Canizales-Rodríguez [2,4], Cinthia Jhovanna Pérez-Martínez [2] and María Teresa Sumaya-Martínez [1]

1  Unidad de Tecnología de Alimentos, Secretaría de Investigación y Posgrado, Universidad Autónoma de Nayarit, Ciudad de la Cultura s/n, Tepic 63000, Mexico; edgar.jimenez@uan.edu.mx (E.I.J.-R.); teresa.sumaya@uan.edu.mx (M.T.S.-M.)
2  Departamento de Ciencias Químico Biológicas, Universidad de Sonora, Blvd. Luis Encinas y Rosales s/n, Hermosillo 83000, Mexico; a216201141@unison.mx (J.A.C.-G.); cesar.otero@unison.mx (C.B.O.-L.); dalila.canizales@unison.mx (D.F.C.-R.); jhovanna.perez@unison.mx (C.J.P.-M.)
3  Universidad Estatal de Sonora. Unidad Académica Navojoa, Blvd. Manlio Fabio Beltrones 810, Col. Bugambilias, Navojoa 85875, Mexico
4  Departamento de Investigación y Posgrado en Alimentos, Universidad de Sonora, Blvd. Luis Encinas y Rosales s/n, Hermosillo 83000, Mexico; saul.ruizcruz@unison.mx
5  Facultad de Ingeniería y Administración, Universidad Nacional de Colombia, Sede Palmira, Palmira 763533, Colombia; almgarzonga@unal.edu.co
6  Carrera en Ingeniería en Innovación Sustentable, Instituto Tecnológico de Estudios Superiores de Zamora, Km. 7, Carretera Zamora-La Piedad, Zamora 59600, Mexico; hebert.jair@uiep.edu.mx
*  Correspondence: victor.ocano@unison.mx (V.M.O.-H.); santiago.valdez@ues.mx (S.V.-H.)

**Abstract:** Fish represent one of the most perishable food groups. Therefore, it is important to find viable alternatives that contribute to the preservation of quality and increase the shelf life of fishery products, and one alternative is to use natural extracts with antimicrobial activity. The objective of this study was to determine the effect of a natural extract prepared with garlic (NGE) on the quality and shelf life of tilapia fillets stored on ice for 18 days. For this purpose, NGE was prepared by homogenizing peeled garlic cloves with distilled water, which were then centrifuged to obtain the extract (NGE); then, the fish fillets were immersed in the extract and were coated in NGE. The fillets with NGE were packed in high-density polyethylene bags and stored in crushed ice for 18 days. The adenosine 5′-triphosphate (ATP) and degradation products, K-value, color, texture, water holding capacity, pH, total mesophilic count, and total volatile bases (TVB-N) were monitored during storage. The ATP content, K-value, pH, total microbial count, and TVB-N changed with respect to ice storage time, and the results between fillets with NGE and control fillets differed. In conclusion, the application of NGE increased the shelf life of fillets stored on ice by 6 days, obtaining a shelf life of 18 days on ice, which shows its potential to improve the utilization of the species.

**Keywords:** extract; garlic; tilapia; freshness; shelf life; quality

**Key Contribution:** One of the main problems involving seafood is its short shelf life during ice storage. For this reason, food technologists are studying different ways to increase its shelf life. In order to solve this problem, this study looks at the effect of a natural extract prepared with garlic on the quality and shelf life of tilapia fillets stored on ice for 18 days.

## 1. Introduction

Fisheries and aquaculture represent an important source of income and employment, and their food products are greatly appreciated by consumers worldwide because of their health benefits, flavor, texture, protein content, types, quality of fatty acids (polyunsaturated), and the variety of vitamins and minerals they provide. However, these products

represent one of the most perishable foods [1], and thus preservation methods should be applied to avoid economic losses and possible negative effects on consumers' health [2–5].

Some authors indicate that the freshness and quality of fishery products decrease immediately after capture and death due to the development of a series of post mortem biochemical changes. These changes can be endogenous or exogenous; the former involves food enzymes, while the latter involves the response to microorganisms' growth and metabolism, as well as storage and handling conditions [6,7].

To reduce spoilage and preserve the shelf life of fishery products, various technologies have been used. Currently, in the fishing industry, several traditional and emerging technologies are being used to preserve fishery products. Traditional technologies include refrigeration, freezing, the use of modified and controlled atmospheres, canning, drying, dehydrating, salting, and smoking [7–9]. Moreover, in the case of new technologies, high pressures, irradiation, bio-preservation, active and intelligent packaging [3,10], and antimicrobial compounds [11], among others, can be used.

Nowadays, there is a growing demand from consumers to purchase fishery products with greater freshness and quality. Thus, several technologies have been developed. For example, the use of natural compounds from plants, animals, and microorganisms are some of the most innovative [12] practices, involving natural extracts, essential oils, or derived compounds [13]. These natural compounds may contain secondary metabolites that slow down or inhibit the development of bacteria, yeasts, and molds, and they contain aldehydes, aliphatic alcohols, phenolic compounds, acids, terpenes, ketones, and isoflavonoids, as well as their precursors, including thymol, carvacrol, eugenol, geraniol, and citral, among others. Until now, antimicrobial compounds have been obtained from rosemary, oregano, onion, green tea, tomato plants, and garlic; however, very few studies have evaluated their usefulness in preserving the quality and extending the shelf life of fish products [8,11,14].

Garlic (*Allium sativum*) is a vegetable with medicinal properties that belongs to the Liliaceae family, and it has biological activity due to its anticarcinogenic effect. Garlic stimulates the immune system, reduces cardiovascular diseases, and has antioxidant and antimicrobial properties [14]. Regarding its antimicrobial effect, garlic extract has been reported to have broad antibacterial activity on *Staphylococcus*, *Klebsiella*, *Clostridium*, *Escherichia*, *Proteus*, *Salmonella*, *Mycobacterium*, and *Helicobacter* [15]. In garlic, the compound that provides the antimicrobial activity is allicin, which is a volatile sulfur-containing molecule with a distinctive odor [16]. Additionally, allicin has been found to act as an antiparasitic, antiviral, and antifungal agent. The literature indicates that the antimicrobial activity of garlic extract is also caused by the presence of hydrophobic compounds such as diallyl polysulfide, ajoene, and vinyldithiin [17]. In the case of tilapia, which is one of the main species cultivated worldwide, to our knowledge no reports such as the one proposed in this study are available. Therefore, the objective of this study is to evaluate the effect of a natural extract made from garlic on the freshness, quality, and shelf life of tilapia (*Oreochromis niloticus*) fillets on ice over 18 days of storage.

## 2. Materials and Methods

### 2.1. Obtaining Tilapia Specimens

The *Oreochromis niloticus* fillets used in this study were purchased commercially from a fishery product retailer in Hermosillo, Sonora, Mexico (average weight and size of $101.87 \pm 14.66$ g and $16.0 \pm 0.41$ cm, respectively). The tilapia fillets were shipped to the Food Research Laboratory at the Universidad de Sonora (UNISON) in Hermosillo, where they were washed with distilled water and ice.

### 2.2. Preparation and Application of the Natural Garlic Extract (NGE)

The natural garlic extract (NGE) preparation process was carried out as follows: 200 g of peeled garlic was homogenized with 800 mL of distilled water in a Molimex (CDMX, México) blender for 2 min. Subsequently, the NGE obtained was centrifuged in a refrigerated centrifuge (SIGMA, Pasadena, CA, USA) at $9000 \times g$ and 4 °C for 30 min. The

supernatant was then collected and cooled to refrigeration temperature (2 °C) prior to its use as a preserving agent. The NGE concentration was of 87.5 g of dry garlic/L of water, and 20 mL were used per kg of fillets (1.75 g of dry garlic per kg of fillets). Its application was carried out as follows: first, the tilapia fillets were washed, drained, and immersed in the NGE for 2 min. Then, they were drained for 5 min, packed in high-density polyethylene bags, and stored in crushed ice for 18 days. At the same time, the tilapia fillets (without NGE) were washed with distilled water, drained, and packed directly into the bags as described above to be used as the control group.

### 2.3. Ice Storage Study

The fillets—with and without NGE—packed in polyethylene bags were subjected to an 18-day ice storage study that consisted of storing the fillets described above in an airtight cooler in alternating beds of ice–fillet–ice. During this time, the determinations of ATP and degradation products, K-value, color, pH, texture, WHC, as well as the total mesophilic count and TVB-N were evaluated at 0, 3, 6, 9, 12, 15, and 18 days of storage. It is important to note that ice was replaced daily and four fillets per sampling day were analyzed for the NGE and control batches, respectively. The analyses performed are described below.

### 2.4. Quality Determinations

#### 2.4.1. Adenosine 5′-Triphosphate (ATP) and Degradation Products

The determination of the ATP and related compound concentration was carried out using high performance liquid chromatography (HPLC) according to the technique described by Ryder [18] that consisted of preparing a homogenate with 3 g of tilapia muscle with 15 mL of 0.6 M perchloric acid at 0 °C. Then, the homogenate was centrifuged for 1 min at 18,000 rpm in an Ultra-Turrax T18 Basic (IKA Works Inc., Wilmington, NC, USA) and centrifuged at 0 °C in a refrigerated Thermo Electron Model IEC-MULTI RF centrifuge (Thermo Fisher Scientific, Asheville, NC, USA) at $5500 \times g$ for 10 min. Subsequently, 7 mL of the supernatant was adjusted to pH 6.5–6. 8 with 1 M KOH, and allowed to stand for 30 min at 0 °C to precipitate the potassium perchlorate and remove it by filtration using Whatman No. 4 paper. Finally, the quantity of the supernatant was increased to 15 mL using distilled water and stored frozen (−80 °C) until quantification.

To determine ATP concentration and related products using HPLC, 20 μL of the neutralized and diluted supernatant was injected into an Agilent 1200 series chromatograph (Agilent, Ratingen, Germany) using a 4.6 mm inner diameter × 250 mm long C18 reverse phase column (ODS Ultrasphere, Beckman Instruments, Inc. Fullerton, CA, USA). A buffer of 0.04 M $KH_2PO_4$ and 0.06 M $K_2HPO_4$ was used as the mobile phase at a flow rate of 1 mL/min. Nucleotides, nucleosides and nitrogenous base were detected at 254 nm with a diode array detector (Agilent 1200 Series Infinity, Agilent Scientific Instruments, Santa Clara, CA, USA).

#### 2.4.2. K-Value (Freshness Index)

The concentrations of adenosine 5′-triphosphate (ATP), adenosine 5′-diphosphate (ADP), adenosine 5′-monophosphate (AMP), inosine 5′-monophosphate (IMP), inosine (HxR), and hypoxanthine (Hx) were used to calculate the K-value. For this purpose, the equation reported by Sagedhal et al. [19] was used, as follows:

K-index (%) = ((HxR + Hx)/(ATP + ADP + AMP + IMP + HxR + Hx)) × 100.

#### 2.4.3. pH

The measurement of the muscle pH was performed using the methodology proposed by Woyewoda et al. [20]. Two grams of tilapia muscle were homogenized with 18 mL of distilled water in an Ultra-Turrax T18 Basic homogenizer (Wilmington, NC, USA) for 1 min at 18,000 rpm. To measure the pH, the electrode of a potentiometer (Orion 420 A, Thermo Electron Corporation, Waltham, MA, USA) was introduced into the homogenate and calibrated daily.

### 2.4.4. Color

The color determination of tilapia fillets from the control lot and with NGE was carried out using tristimulus colorimetry using the CIE L*a*b* color space. For this purpose, a colorimeter (HunterLab MiniScan, Resto, VA, USA) was used, where L*, a*, and b* corresponded to lightness, red-green hue, and yellow-blue hue, respectively. The colorimeter was set in reflectance mode with a reading port opening of 0.5 cm and calibrated according to the manufacturer's specifications.

### 2.4.5. Texture

A texturometer (Shimadzu Model EZ-S, Tokyo, Japan) was used to determine muscle texture. We used a Warner–Bratzler shear cell for this purpose. The maximum shear force (N) required to cut the samples used (dimensions $10 \times 10 \times 20$ mm) was recorded. Thus, the force applied was transverse to the muscle fibers, using a 50 kg compression cell with a head speed of 20 cm/min for the measurement.

### 2.4.6. Water Holding Capacity (WHC)

The WHC of tilapia fillets with NGE and the control was calculated using the technique reported by Cheng et al. [21]. Two grams of fillet was taken and placed in 50 mL centrifuge tubes. The sample was centrifuged for 60 min at $7500 \times g$ at 4 °C in a refrigerated centrifuge (Thermo Fisher Scientific, Asheville, NC, USA). The WHC was calculated as the amount of water loss with relation to the initial content of the sample. The following equation was used for this determination:

$$\text{WHC (\%)} = 100 - ((\text{Wi} - \text{Wf})/\text{Wi} \times 100)$$

where Wi = initial weight, and Wf = final weight after decanting the water from the tube and drying the surface of the muscle.

### 2.4.7. Total Volatile Bases (TVB-N)

The determination of TVB-N was carried out according to the methodology described by Woyewoda et al. [20]. An aliquot of 10 g of each fillet muscle with NGE and the control was transferred to a 1 L ball flask, to which 2 g of magnesium oxide (MgO) and 30 mL of distilled $H_2O$ were added. The mixture was homogenized for 1 min in an Ultra-Turrax (Wilmington, NC, USA). Next, 20 drops of 1-2-3® brand edible vegetable oil were added for defoaming purposes. The TVB-N were distilled for 25 min and collected in 15 mL of 2% boric acid to be later titrated with 0.03 N $H_2SO_4$; similarly, a blank was distilled. Finally, the TVB-N obtained were reported as mg N/100 g of the sample.

### 2.4.8. Aerobic Mesophilic Count

The evaluation of aerobic mesophiles present in tilapia fillets was carried out using the technique described in the Mexican Official Standard "aerobic microorganism plate count" [22]. Ten grams of each fillet with NGE and the control were taken and mixed with 90 mL of 1% (*w/v*) sterile peptonized water using a previously sterilized conventional blender. Subsequently, five serial dilutions were prepared with 1 mL of the largest dilution and mixed with 9 mL of sterile 1% (*w/v*) peptone water. Subsequently, 1 mL of each dilution was manually shaken and placed in a sterile Petri dish, and 20 mL of agar plate count (APC) was poured into the dish. The agar and inoculum were homogenized using the technique described by NOM-092-SSA1-1994 [22]. The plates were incubated in an inverted position at $35 \pm 2$ °C for 48 h. Each of the dilutions was inoculated in triplicate following the same methodology. Finally, between 25 and 250 colony forming units (CFUs) were counted on the dilution. The results were reported as log CFU/g of muscle.

### 2.4.9. Statistical Analysis

The data obtained were analyzed with classical statistical tools such as the mean and standard deviation. Likewise, an analysis of variance and Tukey's multiple range test were

performed. For this purpose, the statistical program NCSS (Version 6.0.22) (Kaysville, UT, USA) [23] was used. For each determination, four replicates were carried out, except for the microbiological determinations, where only two replicates were used. A significance level of 5% was used to analyze the data.

## 3. Results

### 3.1. Effect of NGE on the Post Mortem Biochemistry of Tilapia Muscle

3.1.1. Nucleotide Catabolism

In the muscle of aquatic organisms, ATP degradation represents an important post mortem biochemical change in freshness and quality, where its degradation to Hx has been widely used to monitor freshness in fishery products [24]. Figure 1 shows the average values of ATP concentration and related compounds in tilapia fillets treated with NGE and the control stored for 18 days on ice. The initial ATP values ($0.840 \pm 0.957$ and $1.156 \pm 0.794$ µmol/g) shown in Figure 1 for the fillet with NGE (a) and the control (b) were slightly higher than the 0.26, 0.15, and 0.10 µmol/g of fillet reported by Montoya-Camacho et al. [25], Ocaño-Higuera et al. [26], and Ocaño-Higuera et al. [27] for tilapia (*Oreochromis niloticus*), cazon fish (*Mustelus lunulatus*), and ray fish (*Dayastis brevis*), respectively. However, they were lower than those found in fillets from freshly slaughtered fish (6–12 µmol/g), and this can be attributed to the energy expenditure of the organism during collection and slaughter, as well as the time elapsed between slaughter, filleting, and sampling for analysis. Likewise, Figure 1 shows that IMP was the nucleotide with the highest concentration in tilapia muscle, showing initial values of $4.992 \pm 0.762$ and $4.914 \pm 0.627$ µmol/g for fillets with NGE and the control, respectively. However, the IMP values obtained in this study were lower than the concentrations of 7.0 and 11.3 µmol/g obtained in red tail shad (*Brycon cephalu*) and sierra (*Scomberomorus sierra*) muscles by Batista et al. [28] and Castillo-Yáñez et al. [29], respectively. In the case of nucleotides (ADP and AMP), nucleosides (HxR), and the nitrogenous base (Hx), the initial values were less than 1 µmol/g for both the fillets treated with NGE and the control.

With respect to storage time, only IMP and Hx concentrations showed significant changes ($p < 0.05$), where IMP was reduced to $0.811 \pm 0.119$ and $0.175 \pm 0.172$ µmol/g for fillets with NGE and the control, respectively, at day 18 of ice storage, while Hx increased ($p < 0.05$) from $0.612 \pm 0.173$ and $0.548 \pm 0.0730$ to values of $3.229 \pm 0.173$ and $4.802 \pm 0.768$ µmol/g for fillets with NGE and the control, respectively, also at day 18 of ice storage. This behavior is what normally occurs in fish species during ice storage. The literature indicates that AMP and IMP are the compounds responsible for a sweet taste, which is considered a characteristic of fresh fish [24,30], while the formation and accumulation of HxR and Hx are related to a bitter taste, and consequently to a loss of freshness [31]. In this study, Hx production in tilapia fillets was indicative of the onset of autolytic and bacterial spoilage [32]. In this regard, and with respect to the taste and odor of tilapia fillets with NGE, it is important to highlight that an evaluation carried out by the members of the working group, and not by a sensory panel, of fresh and cooked fillets with NGE detected a slight garlic odor and flavor, which rather than generating negative effects on the perception and acceptance of consumers, provided a better odor and flavor. Therefore, it is considered that NGE has potential for use without affecting consumer acceptance.

Regarding the comparison between NGE and the control, significant differences ($p < 0.05$) were detected in Hx concentrations during the 18 days of ice storage, when the concentration of this nucleoside was lower at the end of storage in fillets with NGE compared to the control. As described above, this result can be related to the onset of microbial growth, and this was further corroborated in the determinations of the K-value, TVB-N, and aerobic mesophilic count.

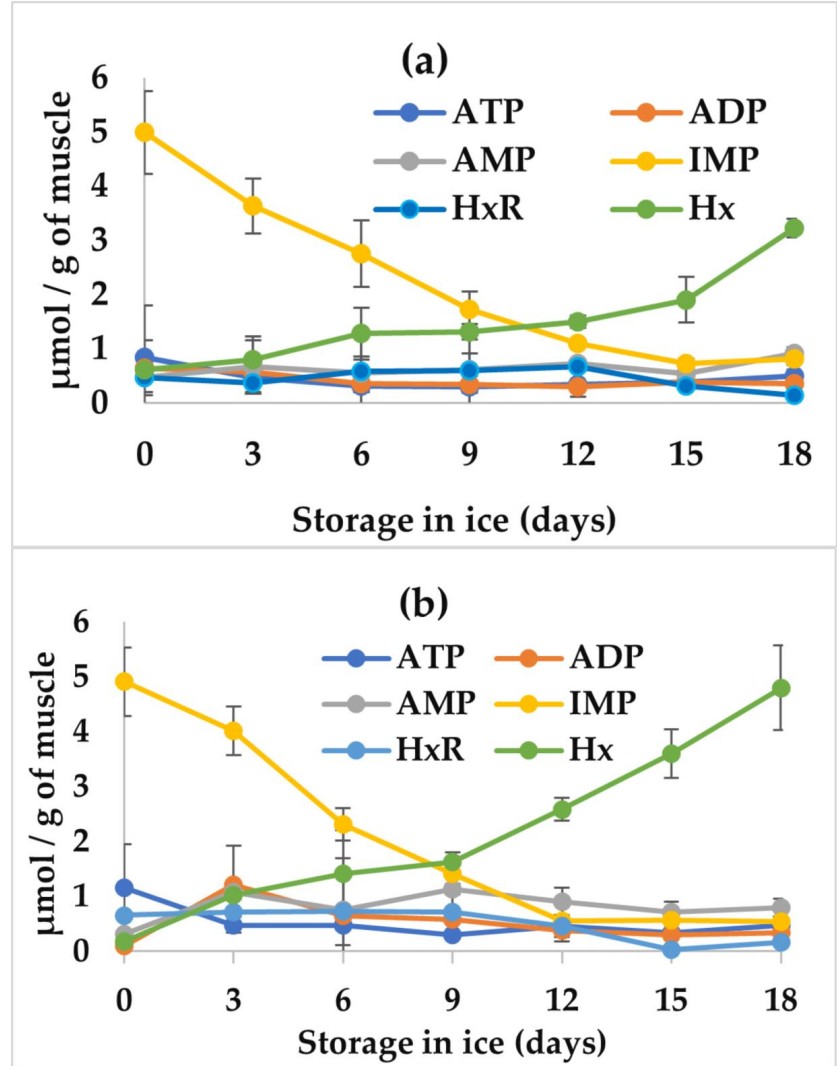

**Figure 1.** Concentration of ATP, ADP, AMP, IMP, HxR, and Hx in fillets of tilapia (*Oreochromis niloticus*) with (**a**) natural garlic extract (NGE) and (**b**) the control, stored for 18 days on ice. The values are the average of n = 4 ± SD (standard deviation). Significant differences ($p < 0.05$) between NGE and the control were found only in Hx.

3.1.2. K-Value

The K-value or freshness index is important to determine the freshness and quality of fishery products. This value is defined as the percentage of the sum of the concentrations of non-phosphorylated ATP degradation products (HxR and Hx) over the total sum of ATP concentrations and degradation products up to Hx [33].

Figure 2 shows the behavior of the K-value in tilapia fillets with NGE and the control stored for 18 days on ice, with initial values of 13.42 and 11.35% for the NGE and control tilapia fillets, respectively. These initial values are similar to those described by Liu et al. [34] and Khalid [35], who reported K-values of 12.00 and 13.80% for tilapia (*Oreochromis niloticus*) and Pacific salmon (*Onchrhynchus nerka*) muscles, respectively. However, the values are lower than those reported by Castillo-Yáñez et al. [29], who obtained an initial value of 21.23% for sierra (*Scomberomorus sierra*) muscle.

With respect to tilapia fillet ice storage, at day 18 a linear and significant increase ($p < 0.05$) was observed in the determination of the K-value, reaching final values of 56.51 and 69.61% for tilapia fillets treated with NGE and the control, respectively. These values are higher than 39.5 ± 2.6% found by Montoya-Camacho et al. [25] in tilapia fillets stored for 20 days at 0 °C. This difference may be due to the fact that the fillets in this study were

acquired commercially, while those used by Montoya-Camacho et al. [25] were acquired freshly slaughtered directly from an aquaculture company. Moreover, the K-values of this study are similar to the 58.9% described by Ocaño-Higuera et al. [26] in cazon fish (*Mustelus lunulatus*) stored for 18 days on ice. Furthermore, the values obtained are lower than those found by Özoğul et al. [36] and Castillo-Yáñez [29], who reported a final value of 90.0 and 80.6% for turbot (*Scophthalmus maximus*) and sierra (*Scomberomorus sierra*) fillets, respectively. Finally, a significant effect ($p < 0.05$) was found between tilapia fillets with NGE and the control, which showed that its application extends the shelf life of tilapia fillets, and this may be a consequence of the antimicrobial and antioxidant properties of garlic [37].

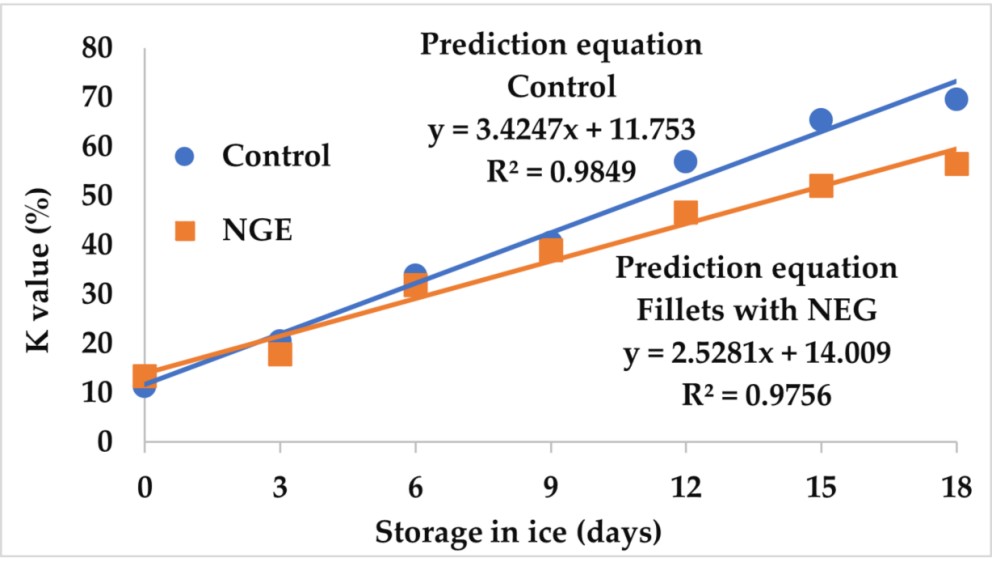

**Figure 2.** Behavior of the K-value in tilapia fillets with natural garlic extract (NGE) and the control stored on ice for 18 days. Values are the mean of n = 4 ± SD (standard deviation).

Saito et al. [38] reported a quality classification based on the K-value, for which they indicated that fishery products with values below 20% were very fresh, values below 50% were moderately fresh, and values above 70% were not fresh and should not be consumed. Based on this classification, tilapia fillets from the control at day 18 were almost in the not fresh category (69.60%), while those with NGE showed a lower edible quality than those moderately fresh. However, it should be emphasized that to use the aforementioned classification, it is necessary to take into account that the K-index is a parameter that depends on the species. Thus, it should be calculated for each fish species.

### 3.2. Effect of the NGE on Tilapia Muscle Physicochemical Quality
#### 3.2.1. pH

In general, the pH of live fish immediately after slaughter is close to 7.0, but this value decreases rapidly immediately after death, which occurs as glycogen is converted to lactic acid and fish go through rigor mortis. Most fish species reach post mortem pH values from 6.0 to 6.8 [39]. This measurement is of great importance when judging the freshness and quality of fishery products [40]. Figure 3 shows the behavior of the pH in the fillets stored for 18 days on ice, where the initial pH values of 7.04 ± 0.143 and 7.24 ± 0.028 can be observed for the fillets with NGE and the control, respectively. These initial values are higher than those reported by Tomé et al. [41] and Montoya Camacho et al. [25], who obtained initial pH values of 6.48 and 6.77, respectively, in tilapia (*Oreochromis* spp.) muscle stored on ice. These differences may be due to species, diet, muscle type, season of the year, and activity level, as well as to stress during capture [26–28,41,42].

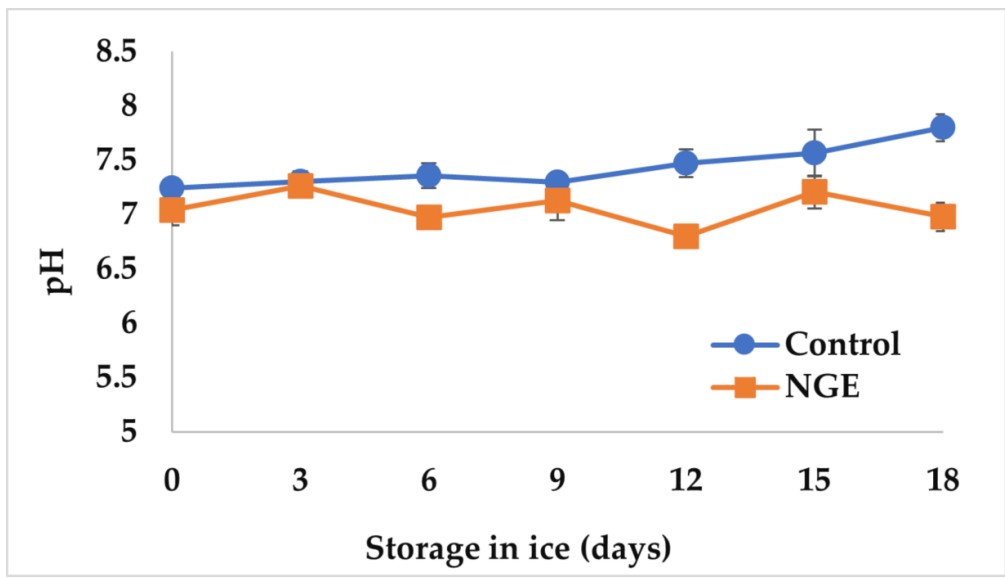

**Figure 3.** pH values obtained in tilapia fillets (*Oreochromis niloticus*) with NGE and the control stored on ice for 18 days. Values are the mean of n = 4 ± SD (standard deviation).

Furthermore, according to Figure 3, the pH increased significantly ($p < 0.05$) only in the control fillets, finding a value of 7.8 ± 0.12 at day 18 of ice storage. These values are higher than those described by Khalafalla et al. [39], Attouchi and Sadok [43], and Tomé et al. [41], who reported final pH values of 7.2, 6.80, and 6.73 for the muscles of *Oreochromis niloticus*, *Spaurus aurata*, and *Oreochromis* spp., respectively. The increase in pH observed in the control fillets during storage can be related mainly to the generation of volatile compounds such as amines produced through autolytic means (tyramine, putrescine, histamine, and cadaverine) and ammonia [44], as well as to the degradation of free amino acids by bacterial action [34]. Moreover, a significant increase ($p < 0.05$) in pH was observed in the tilapia fillets with NGE; however, in the control fillets, the pH did not increase ($p > 0.05$) during storage on ice. This result could be caused by the presence of antimicrobial compounds in garlic, in which some organosulfur compounds are found, such as allyl sulfides, ajoene, and allicin [45].

Regarding the effect of NGE application on the pH of tilapia fillets, significant differences ($p < 0.05$) were observed in this parameter between fillets treated with the extract and the control. Fillets with NGE showed a lower pH than the control fillet. This result may be due to the presence of some organic acids found in garlic bulbs, including pyruvic, malic, and citric acids [46].

### 3.2.2. Color

Color is a parameter used to evaluate the quality of fishery products, and it especially influences consumer acceptance for these products [26]. Figure 4 shows the values obtained for the parameters L*, a*, and b* in the tilapia fillets with NGE and the control stored for 18 days on ice. For the L* parameter, initial values of 62.02 ± 1.78 and 60.04 ± 3.10 were observed for fillets treated with the extract and the control, respectively. The initial values are similar to the 62.3 ± 1.5 reported by Montoya-Camacho et al. [25] for tilapia fillets at the beginning of storage. With respect to storage, only the control fillets showed a slight significant increase ($p < 0.05$), reaching a value of 65.40 ± 4.74 at 18 days. This increase could be caused by the loss of fluids through exudation that occurs in the fillet; this generates a watery appearance on the fillet surface and, thus, an increase in brightness [32]. Regarding the effect of NGE, no differences were found ($p > 0.05$) compared with the control.

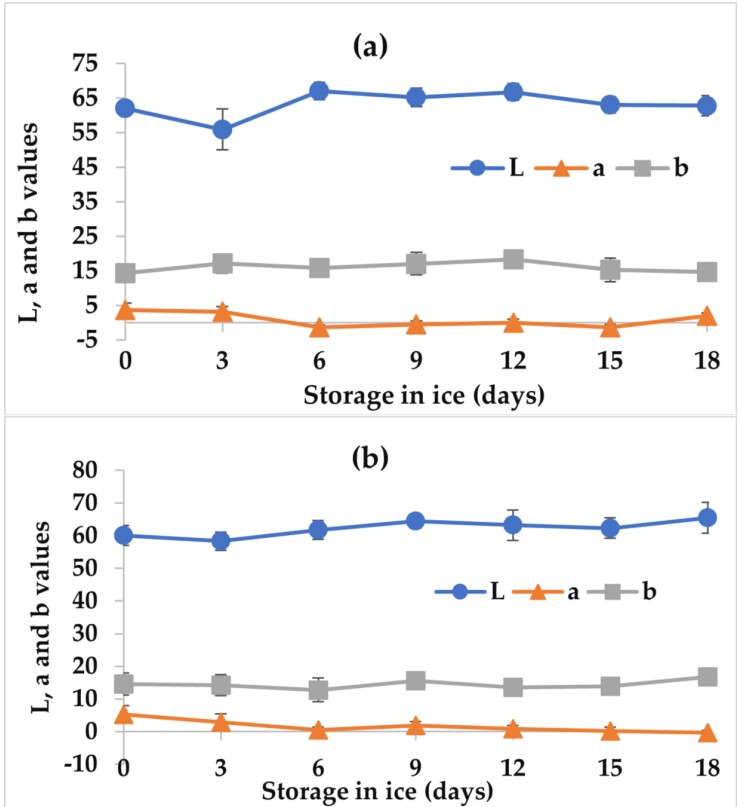

**Figure 4.** Change of color (L*, a*, and b* values) in tilapia fillets (*Oreochromis niloticus*) with (**a**) NGE and (**b**) the control stored for 18 days on ice. The values are the average of n = 4 ± SD (standard deviation).

Moreover, in parameter a*, tilapia fillets with NGE and the control showed a slightly significant reduction ($p < 0.05$) with respect to storage time, obtaining values of 1.94 ± 0.77 and −0.36 ± 0.64 at day 18 of ice storage, respectively. This decrease could be attributed to the fact that some sarcoplasmic proteins, such as myoglobin and hemoglobin, are solubilized in the water lost by the fillet during storage, causing the red color to decrease [32]. In the b* parameter, the values did not change significantly ($p > 0.05$) with respect to storage time in ice nor with the NGE application. Consequently, the color of tilapia fillets with the extract and the control showed colorations towards orange-yellow in the chromatic sphere, recording hue values of 82.46 and 91.23, respectively.

### 3.2.3. Texture and Water Holding Capacity (WHC)

Texture evaluation is an important parameter to assess quality in fresh products [47,48]. Muscle smoothness is modified by several factors such as the degradation of Z-discs, the destruction of connectin, the dissociation of the actin–myosin complex, and by collagen denaturation. Texture is also affected by endogenous proteases that degrade myofibrillar proteins. The latter seems to be the main cause of muscle softening, since some of them are fully active at a post mortem pH between 5.5 and 6.5 [49].

Figure 5a shows texture behavior evaluated by the sheer force technique in tilapia fillets with NGE and the control during 18 days of storage on ice. This figure shows that the initial values were 5.80 ± 1.01 and 6.54 ± 0.48 N for the fillets treated with the extract and the control, respectively. These values are similar to the initial value of 6.52 ± 0.22 N reported by Montoya-Camacho et al. [25] for tilapia (*Oreochromis niloticus*) fillets. This quality parameter was not affected by either ice storage time or NGE application ($p > 0.05$). Therefore, since no significant differences ($p > 0.05$) were observed in this determination of shear force between storage time and treatments, and the results suggest that the autolytic

enzymes responsible for muscle fiber degradation did not affect the muscle integrity of the fillets.

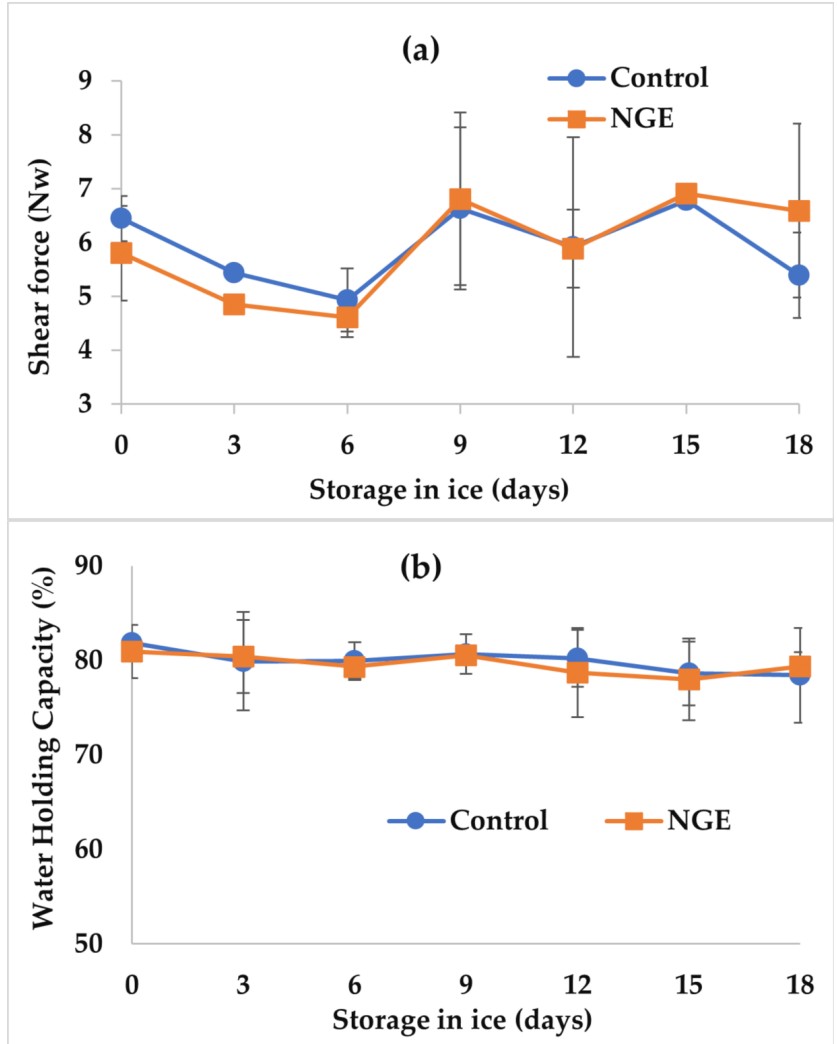

**Figure 5.** Changes in texture (**a**) and water holding capacity (**b**) in tilapia fillets with natural garlic extract (NGE) and the control stored on ice for 18 days. Values are the mean of n = 4 ± SD (standard deviation).

The WHC is the ability of meat to retain its own water during the application of external forces [50]. A loss in its value is mainly due to the denaturation (hydrolysis and/or aggregation) that occurs in the myofibrillar protein of meat during storage in ice [26], which is related to the loss of muscle texture [51,52].

Figure 5b illustrates the behavior of WHC in tilapia fillets treated with NGE and the control during 18 days of ice storage. The figure shows that the initial WHC values were 80.92 ± 2.81 and 81.84 ± 0.35% for fillets with the extract and the control, respectively. These initial values are lower than those described by Montoya-Camacho et al. [25], who obtained initial WHC values of 95.3 ± 0.4% in tilapia (*Oreochromis* spp.) fillets stored on ice. These differences can be attributed to diet, stress during capture, species, season of the year, or type of muscle studied, as well as activity level [26–28,41,42]. With respect to ice storage, no significant changes ($p > 0.05$) were observed in the fillets with NGE and the control, reaching values of 79.36 ± 0.52 and 78.43 ± 5.00%, respectively. These values are lower than those described by Coronado and Moreno ([32], who reported a final value of 88.5% for tilapia (*Oreochromis niloticus*) muscle. Likewise, no significant differences ($p > 0.05$) were found between the control and the fillets with NGE.

### 3.2.4. Total Volatile Bases (TVB-N)

The determination of TVB-N is widely used to evaluate the quality of fishery products [53]. Figure 6a shows the behavior of TVB-N in fillets treated with NGE and the control stored for 18 days on ice. The figure shows that the initial values were $22.13 \pm 1.29$ and $22.82 \pm 1.70$ mg N/100 g of muscle for the fillets with the extract and the control, respectively. These values are similar to the $26.2 \pm 1.6$ mg N/100 g reported by Montoya-Camacho et al. [25] and are higher than the 6.5 and 10.2 mg N/100 g for tilapia (*Oreochromis niloticus*) muscle described by Liu et al. [34] and Pankyamma et al. [54], respectively. Regarding storage time, a significant increase ($p < 0.05$) in TVB-N values was observed, reaching up to $34.76 \pm 10.60$ and $60.21 \pm 10.56$ mg N/100 g in the tilapia fillets with NGE and the control at day 18, respectively. In addition, significant differences ($p < 0.05$) were found between the control and the fillets with the extract. In the case of fishery products, a maximum allowable value of 30 mg N/100 g of muscle is used for a product fit for human consumption [55]. This value was reached at 12 days in the control fillet and at 18 days in the fillet with NGE. It is important to note that for these days of storage, the pH and microbial load values also increased. Therefore, the TVB-N content increased due to the growth of spoilage microorganisms that are capable of generating volatile compounds characteristic of spoilage. This result indicated that NGE increased the shelf life of tilapia fillets by six days compared to the control batch.

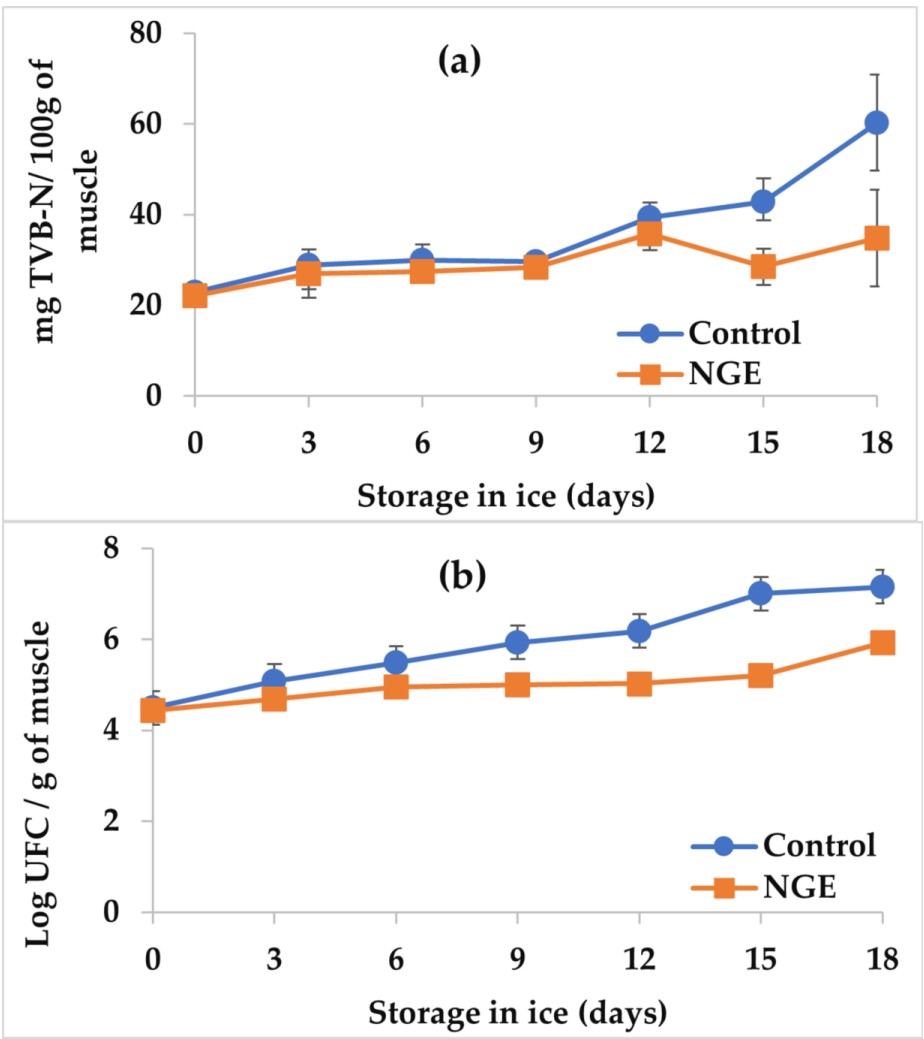

**Figure 6.** Changes in TVB-N (**a**) and the bacterial viable counts (**b**) in tilapia fillets with natural garlic extract (NGE) and the control stored on ice for 18 days. Values are the mean of n = 4 $\pm$ SD (standard deviation).

### 3.3. Effect on the Microbiological Quality

Currently, the fishing industry is suffering from great economic losses due to the rapid deterioration of this group of foods, which is a consequence of the microbiota they present, where they mostly possess Gram-negative spoilage bacteria [44]. In general, the microbial load of fishery products is closely related to the initial level of bacterial contamination and the conditions of the aquatic environment (salinity, light availability, oxygen levels, density, pH, etc.), as well as to the preservation method applied (cold, heat, reduction in moisture content, biopreservation, etc.) that influences the growth and survival of bacteria [56].

Figure 6b shows the behavior of the mesophilic microbial load in tilapia fillets treated with NGE and the control during 18 days of storage on ice, where values of 4.43 and 4.49 log CFU/g were obtained for the fillets with the extract and the control, respectively. This value is higher than that reported by Coronado and Moreno [52], who found an initial value of 3.2 log CFU/g for tilapia fillets (*Oreochromis niloticus*). Moreover, a significant increase ($p < 0.05$) in the microbial count was observed with respect to storage time, where values of 5.92 and 7.15 log CFU/g were obtained for the fillets with the extract and the control, respectively. Likewise, significant differences ($p < 0.05$) were observed between the fillets with NGE and the control, and this showed the antimicrobial power of the latter. Similar studies have shown the antimicrobial effect of garlic on tilapia fillets. For example, Tawfik et al. [57] and Kirrella et al. [58] used essential oils from this vegetable. Studies report that the capacity of garlic or the extracts obtained from it is mainly due to allicin, ajoene, and various aliphatic sulfides [45]. Unlike these studies, the present work has the advantage that a simple garlic extract was used and it was easy to apply to fish fillets.

The Mexican Official Standard (NOM-027-SSA1-1993) [59] establishes 7 log CFU/g or 10,000,000 CFU/g of microorganisms as the maximum permitted value for fresh and/or refrigerated fish fit for human consumption. Based on the results obtained in this determination, the control fillets had an edible quality up to day 12 of storage, while the fillets with the extract continued to be of edible quality even at day 18 of storage on ice. These results agree with those described above for the K-index, TVB-N, and the increase in hypoxanthine content at day 18 of storage.

### 4. Conclusions

The determinations of the K-index, total mesophilic count, total volatile bases, and pH showed their usefulness in evaluating the antibacterial effect of NGE, the application of which, by immersion, in commercial tilapia fillets increased the shelf life of the fillets by six days compared to the control batch. In general, the results of this study showed the potential of NGE to extend the shelf life of tilapia fillets, which could consequently lead to a greater utilization of this species, and to less waste. In addition, this extract may be applied to other fishery species to extend the shelf life and reduce post-harvest losses or wastage.

**Author Contributions:** V.M.O.-H., S.V.-H. and E.I.J.-R. designed the study. J.A.C.-G., C.B.O.-L., D.F.C.-R. and C.J.P.-M. performed formal analyses of the manuscript. V.M.O.-H., S.R.-C., A.M.G.-G., H.J.B.-C. and M.T.S.-M. wrote the original draft. All authors contributed to writing, reviewing, and editing the manuscript. All authors contributed to the research and approved the submitted version. All authors have read and agreed to the published version of the manuscript.

**Funding:** This research study was J.A.C.G.'s undergraduate thesis and was supported by the University of Sonora (No research project). Some members of the Red Temática de Bioproductos y Bioprocesos participated in this study.

**Institutional Review Board Statement:** Not applicable.

**Informed Consent Statement:** Not applicable.

**Data Availability Statement:** The data presented in this study are available upon request from the corresponding author.

**Conflicts of Interest:** The authors declare no conflict of interest.

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
