# Peer review of "Quality Improvement and Shelf-Life Extension of Iced Nile Tilapia Fillets Using Natural Garlic Extract"

_fishes, doi:10.3390/fishes8060325_

Round 1
Reviewer 1 Report
Authors studied the effect of water extract from garlic on the freshness and quality of tilapia fillet stored in ice, to increase the preservability. The manuscript is well written and kind to read. No major errors were found in the experimental design and results. The results, self-stability of fish fillet can be improved through the NGE treatment, would be expected to have a positive industrial impact.
However, the following issues should be reconsidered and revised.
1. Please reconsider the title.
2. In the abstract, please add the following contents briefly: explanation of NGE, NGE treatment method and storage condition, and the actual shelf-life with the values of critical freshness indicator.
3. Line 85, please delete the sentence, there are several reports that address positive effect of garlic extract on the self- stability of fish fillets, including tilapia.
4. In the materials and method, did authors investigate the major compounds capable for antimicrobial activity in NGE?
5. Was the core temperatures of fillet measured during storage?
6. Line 99, please give the spin speed.
7. Line 101, ‘decanted’ revise by ‘collected’
8. Line 208, ‘biochemical change in quality’ revise by ‘biochemical change in freshness and quality’
9. Lines 211-212, please represent!
10. Line 219, please provide the full name of IMP, as well as, HxR and Hx, at the first time.
11. Please provide statistical significance between NGE and control in all figures, except for figure 2.
12. Did authors measure the moisture content of the fillet? The increase in the moisture content is expected after NGE treatment, it affected on the shear force and water holding capacity.
Minor editing of English language required.
Author Response
Reviewer 1
Reviewer comments
Authors studied the effect of water extract from garlic on the freshness and quality of tilapia fillet stored in ice, to increase the preservability. The manuscript is well written and kind to read. No major errors were found in the experimental design and results. The results, self-stability of fish fillet can be improved through the NGE treatment, would be expected to have a positive industrial impact.
However, the following issues should be reconsidered and revised.
- Please reconsider the title.
Use of garlic (Allium sativum) extract to preserve freshness and quality and extend shelf life of tilapia (Oreochromis niloticus) fillets stored in ice
Authors' response
Dear reviewer, we greatly appreciate your feedback. The title was changed to:
Quality improvement and shelf-life extension of iced Nile tilapia fillets using natural garlic extract
Reviewer comments
- In the abstract, please add the following contents briefly: explanation of NGE, NGE treatment method and storage condition, and the actual shelf-life with the values of critical freshness indicator.
Authors' response
Dear reviewer, all suggestions were included in the abstract. Undoubtedly, their comments substantially improved the summary, which was as follows:
Abstract: Fish represent one of the most perishable food groups. Therefore, it is important to find viable alternatives that contribute to the preservation of quality and increase the shelf life of fishery products, and one alternative is to use natural extracts with antimicrobial activity. The objective of this study was to determine the effect of a natural extract prepared with garlic (NGE) on the quality and shelf life of tilapia fillets stored on ice for 18 days. For this purpose, NGE was prepared by homogenizing peeled garlic cloves with distilled water, which were then centrifuged to obtain the extract (NGE); then, the fish fillets were immersed in the extract and were coated in NGE. The fillets with NGE were packed in high-density polyethylene bags and stored in crushed ice for 18 days. The adenosine 5'-triphosphate (ATP) and degradation products, K-value, color, texture, water holding capacity, pH, total mesophilic count, and total volatile bases (TVB-N) were monitored during storage. The ATP content, K-value, pH, total microbial count, and TVB-N changed with respect to ice storage time, and the results between fillets with NGE and control fillets differed. In conclusion, the application of NGE increased the shelf life of fillets stored on ice by 6 days, obtaining a shelf life of 18 days on ice, which shows its potential to improve the utilization of the species.
Reviewer comments
- Line 85, please delete the sentence, there are several reports that address positive effect of garlic extract on the self- stability of fish fillets, including tilapia.
Authors' response
The statement was deleted.
Reviewer comments
- In the materials and method, did authors investigate the major compounds capable for antimicrobial activity in NGE?
Authors' response
A very interesting question. For the present manuscript, the compounds that have antimicrobial activity were not determined, because only the effect of garlic extract was evaluated. However, in the near future, we will determine the compounds that have antimicrobial activity, as well as test them against spoilage and pathogenic bacteria, as well as establish the minimum inhibitory and bactericidal concentration, which we are already in the process of determining in order to conclude with a master's thesis and the writing of the respective manuscript.
Reviewer comments
- Was the core temperatures of fillet measured during storage?
Authors' response
We appreciate your question. The answer is that the core temperature of the fillet was not measured during storage. However, it is considered that the temperature was 0°C, since the fillets were always stored with plenty of ice made with purified water.
Reviewer comments
- Line 99, please give the spin speed.
Authors' response
We missed including the spin speed. We appreciate your comment. The spin speed was included in the statement.
Line 101 Subsequently, the NGE obtained was centrifuged in a refrigerated centrifuge (SIGMA, Pasadena, USA) at 9000 x g and 4°C for 30 min.
Reviewer comments
- Line 101, ‘decanted’ revise by ‘collected’
Authors' response
Line 102: Decanted was changed by collected
Reviewer comments
- Line 208, ‘biochemical change in quality’ revise by ‘biochemical change in freshness and quality’
Authors' response
Linea 211: biochemical change in quality’ was changed by ‘biochemical change in freshness and quality’
Reviewer comments
- Lines 211-212, please represent!
Authors' response
Items a) and b) in Figure 1 were included.
Lines 214-216
The initial ATP values (0.840 ± 0.957 and 1.156 ± 0.794 µmol/g) shown in Figure 1 for the fillet with NGE (a) and the control (b) are slightly higher than the 0.26, 0.15, and 0.10 µmol/g of fillet reported by Montoya-Camacho et al. [25],
Reviewer comments
- Line 219, please provide the full name of IMP, as well as, HxR and Hx, at the first time.
Authors' response
Lines 142 and 143 describe the abbreviations for IMP, HxR and Hx.
Reviewer comments
- Please provide statistical significance between NGE and control in all figures, except for figure 2.
Authors' response
Statistical significance between NGE and control was included in Figures
Reviewer comments
- Did authors measure the moisture content of the fillet? The increase in the moisture content is expected after NGE treatment, it affected on the shear force and water holding capacity.
Authors' response
Thank you very much for your comment. We did not determine the moisture content in the fillets with NGE. As you indicate, the immersion of the fillets could modify the moisture content and with it the WHC and texture results, however, it is important to highlight that although the application of the NGE was by immersion, this was superficial and that it was left to drain for 5 min to eliminate the excess NGE. In addition, no significant changes were observed in texture or WHC with respect to the fillets with NGE and the control.
Additionally, the English of the manuscript was reviewed by
the FIshes English language editing team.

Reviewer 2 Report
Manuscript ID: fishes-2414932
Use of garlic (Allium sativum) extract to preserve freshness and quality and extend shelf life of tilapia (Oreochromis niloticus) fillets stored in ice
The authors present an article focusing on the potential use of garlic to contain the microbial contamination of tilapia fillets stored in ice, therefore, to extend their shelf life. As reported by the authors in the introduction, the consumers are interested in novel and natural antimicrobial compounds used to replace the common chemical additives.
However, this practice could negatively affect the odour and flavour of fillets and this aspect is not taken into consideration by the authors. This point is missing, and it represents the main weakness of the manuscript.
In my opinion, the manuscript is focused on a very interesting topic, however, some integrations and revisions are needed.
Some revisions are suggested and listed below.
Overall, the text needs a moderate English check, some grammatical and typo errors are present.
Some doubts and missing information are related the section “Materials and Methods”:
- Line 104: please, specify the concentration of NGE (g powder/L water) and the quantity of water used (ratio between total weight of fillets and water).
- The panel test is missing, and I believe it could be an essential point to allow considering garlic as a potential compound to be added in food treatments.
Regards the results, please specify that the paragraph is also focus on discussion.
- Figure 1: I suggest creating 6 graphs and, in each of them, compare the ctr and treated samples, making it easier to understand the differences.
- Figure 3: please, add statistic.
- Figure 4: : I suggest creating 3 graphs and, in each of them, compare the ctr and treated samples, making it easier to understand the differences.
-
I would like to suggest to:
- Insert more references in the introduction; for example, in lines 69-71;
- Check the abbreviations, entering the full name when you first mention them in the manuscript.

Overall, the text needs a moderate English check, some grammatical and typo errors are present.
Reviewer 3 Report
Authors studied the effect of garlic extract on shelf-life status and quality and of chill-stored tilapia fillets. Results indicated that the use of GE extended by 6 days the shelf life of fillets. In general, the work is interesting and the aim is clear. However, there are several issues that have to be carefully addressed by authors. Furthermore, English editing is strongly recommended, since there are several parts throughout the manuscript that are difficult to be followed.
-L23-24. Please revise.
-L46. When you refer to such aspects, you should include some important refs e.g., Gram & Huss, (1996); Gram & Dalgaard, (2002); Zhuang et al., (2021); Anagnostopoulos et al., (2022).
-L46-47. Please add some refs.
-L54-55. Please revise.
-L97-107. Why authors did not apply different concentrations of GE?
-L111-115. What about sensory evaluation during storage? This would be very useful for such kind of experiments.
-L184-196. Why authors did not study some other microbial groups (e.g., Pseudomonas, H2S producing-bacteria, Enterobacteriaceae, etc.) rather than just APC? Such information is very important.
-L207-209. Section 3 should be revised to “results and discussion”.
-L285-287. Indeed. Thus, sensory evaluation throughout storage, would be very helpful to determine and compare the shelf-life of the studied products and is totally missing from the present work.
-L383. Please revise.
-L415-417. Please revise.
-L419. “the conditions of the aquatic environment” what does it mean?
-L419-420. Please provide a thorough discussion.
-L427-428. This finding indicates a retarded growth of SSOs and/or a potential different dominant microbiota profile. Thus, as I mentioned above, studying other microbial groups (e.g., Pseudomonas, H2S producing-bacteria, Enterobacteriaceae, etc.) would be very interesting to check this hypothesis.
-L430-434. This is just an indication. The population of spoilage microorganisms at the time of rejection is called microbial spoilage level and varies from log 6 to log 9 cfu/g, because it is depended on spoilage potential and activity of spoilage microorganisms, fish species, storage conditions etc. There is a plenty of published works in seafood spoilage, supporting the aforementioned. Thus, this paragraph should be revised accordingly.
-L442-449. Conclusions should clearly highlight the following. Which is the usefulness of the present work? How important are the findings? How scientific/industry communities would be benefited by the findings of this work? What are the next steps?
English editing is strongly recommended, since there are several parts throughout the manuscript that are difficult to be followed.
Reviewer 4 Report
Manuscript ID: fishes-2414932-peer-review-v1-1
Use of garlic (Allium sativum) extract to preserve freshness and quality and extend shelf life of tilapia (Oreochromis niloticus) fillets stored in ice
The present review provides an overview on recent strategies based on natural compounds addition to marine species to improve quality of chilled product. In particular this work aims to determine the effect of a natural extract prepared with garlic (NGE) on quality and shelf life of tilapia fillets stored on ice for 18 days. The topic is very interesting and important topic in the field of food hygiene and technology. The manuscript was well written and the concepts well explanined.
I believe that this manuscript does not need big changes and published after minor revision
The minor revision concern an update of references because literature is not always complete;
Specific suggestions
Lines 56 – 58: traditional technologies…
· Please add: Aponte M., Anastasio A., Marrone R., Mercogliano R., Peruzy M.F., Murru N. 2018. Impact of gaseous ozone coupled to passive refrigeration system to maximize shelf-life and quality of four different fresh fish products. LWT Volume 93, Pages 412 – 419
Lines 64 natural extracts
· please add: A. Anastasio, R. Marrone, C. Chirollo, G. Smaldone, M. Attouchi, P. Adamo, S. Sadok, T. Pepe (2014). Swordfish steaks vacuum-packed with Rosmarinus officinalis. ITALIAN JOURNAL OF FOOD SCIENCE, vol. 26, p. 390-397, ISSN: 1120-1770
Reviewer 5 Report
Introduction: Authors do present as innovative preservation methods, research that has been reviewed almost 15 years ago (Cortesi et al., 2009) and moreover methods with very limited practical use ever since.
Authors should outline the real novelty, the practical use of their preservation method. Garlic has already used as
Materials and methods: Authors purchase fish from a retailer, meaning that they have no clue about the freshness and the prior handling conditions after slaughtering fish. What was the origin of the fish? What was the initial freshness of the fish What is the purpose of washing fish with distilled water and ice? Furthermore, this is not a custom practice for fish in commercial conditions, is it?
M&M in respect to preservation, treatment and analyses are adequately described.
Results & discussion: results should not be presented in duplicate in text and figures and tables. In some points authors compare the individual ATP breakdown products concentrations (or the K-value) to those of other irrelevant species from different studies (e.g. L 222, 223, 258, 260, 271-275). What is the meaning of doing so, and what is the point they want to make? Same stands for other physicochemical parameters that change with storage. Authors should focus on make comparisons with other studies that occur in
The differences in final K-values for the same fish species can be also attributed to different storage conditions (except the origin of fish).
Discussion is really poor. Discussion should be largely re-written, focusing on the effect of garlic on extending shelf life and microbial inhibition in fresh fish.
General comments:
A lot of occurring literature on the use of garlic products in fresh fish product shelf life preservation is completely neglected. Authors should make a good literature search and review before resubmitting this manuscript to any
A thorough English editing is absolutely necessary, since numerous language mistakes (both grammatical and expression ones) occur throughout the document.
Specific comments
L23: Fish represent
L23-24: “viable alternatives that contribute to preserve initial attributes is important” English –does not make sense (what is viable alternatives? Alternatives to what? what initial attributes?).
L26: on the quality
L45: “these products represent one of the most perishable food”- English
L49: “the development of a series of postmortem biochemical change”- English
L227, L237: “significant differences (p > 0.05)” significant differences stand for p lower than 0.05 (not higher)
A thorough English editing is absolutely necessary, since numerous language mistakes (both grammatical and expression ones) occur throughout the document.
I started pointing out some mistakes in the specific comments, but there were too many and I gave up.
Round 2
Reviewer 1 Report
The MS has been well revised and is suitable for publication in Fishes.
Reviewer 3 Report
The revised manuscript has been strongly improved and authors addressed the majority of my remarks. I have no further comments.